# Lithium-Sodium Separation by a Lithium Composite Membrane Used in Electrodialysis Process: Concept Validation

**DOI:** 10.3390/membranes12020244

**Published:** 2022-02-21

**Authors:** Takoua Ounissi, Rihab Belhadj Ammar, Christian Larchet, Lobna Chaabane, Lassaad Baklouti, Lasâad Dammak, Emna Selmane Bel Hadj Hmida

**Affiliations:** 1Laboratoire de Chimie Analytique et d’Électrochimie, Département de Chimie, Faculté des Sciences de Tunis, Campus Universitaire, Tunis 2092, Tunisia; ounissi.takoua@gmail.com (T.O.); belhadj.rihab25@gmail.com (R.B.A.); emnaselmane@gmail.com (E.S.B.H.H.); 2Université Paris-Est Créteil, CNRS, ICMPE, UMR 7182, 2 Rue Henri Dunant, 94320 Thiais, France; larchet@u-pec.fr (C.L.); lobna.chaabane@u-pec.fr (L.C.); dammak@u-pec.fr (L.D.); 3Department of Chemistry, College of Sciences and Arts at Ar Rass, Qassim University, Ar Rass 51921, Saudi Arabia

**Keywords:** electrodialysis, lithium selective membrane, lithium-sodium separation, membrane selectivity, lithium recovery rate

## Abstract

The recent expansion of global Lithium Ion Battery (LIBs) production has generated a significant stress on the lithium demand. One of the means to produce this element is its extraction from different aqueous sources (salars, geothermal water etc.). However, the presence of other mono- and divalent cations makes this extraction relatively complex. Herein, we propose lithium-sodium separation by an electrodialysis (ED) process using a Lithium Composite Membrane (LCM), whose effectiveness was previously demonstrated by a Diffusion Dialysis process (previous work). LCM performances in terms of lithium Recovery Ratio (RR(Li^+^)) and Selectivity (S(Li/Na)) were investigated using different Li^+^/Na^+^ reconstituted solutions and two ED cells: a two-compartment cell was chosen for its simplicity, and a four-compartment one was selected for its potential to isolate the redox reactions at the electrodes. We demonstrated that the four-compartment cell use was advantageous since it provided membrane protection from protons and gases generated by the electrodes but that membrane selectivity was negatively affected. The impact of the applied current density and the concentration ratio of Na^+^ and Li^+^ in the feed compartment ([Na^+^]_F_/[Li^+^]_F_) were tested using the four-compartment cell. We showed that increasing the current density led to an improvement of RR(Li^+^) but to a reduction in the LCM selectivity towards Li^+^. Increasing the [Na^+^]_F_/[Li^+^]_F_ ratios to 10 had a positive effect on the membrane performance. However, for high values of this ratio, both RR(Li^+^) and S(Li/Na) decreased. The optimal results were obtained at [Na^+^]_F_/[Li^+^]_F_ near 10, where we succeeded in extracting more than 10% of the initial Li^+^ concentration with a selectivity value around 112 after 4 h of ED experiment at 0.5 mA·cm^−2^. Thus, we can objectively estimate that the concept of this selective extraction of Li^+^ from a mixture even when concentrated in Na^+^ using an ED process was validated.

## 1. Introduction

As a response to the cost-effective energy storage system, Lithium-Ions Batteries (LIBs) have emerged as the most promising solution [1,2,3]. In fact, as the lightest metal of the periodic table, lithium has the highest electrochemical potential and the highest energy density by weight of all metals [4]. Such characteristics make it the most popular commodity of modern life for supplying batteries. In addition to LIBs, which are estimated to be around 71% of the lithium end-used in 2021 [5], lithium serves widely for glass and ceramics production, lubricating greases, polymer production, air treatment etc.

The main primary natural lithium resources are brines (salars, seawater, geothermal water etc.) and minerals (spodumene, lepidolite and petalite) [6]. Spent LIBs can also be considered as secondary lithium resources because of the extensive use in the last decades, although exploitation remains very limited [7,8].

Thus, this rapid development of LIBs has made the lithium supply a strategic and global issue that has led both academic and industrial research to explore effective and selective lithium recovery from its resources. Nevertheless, its extraction from brines is advantageous compared to minerals due to its availability in solution.

For lithium recovery from its aqueous resources, generally as LiCl or LiCO_3_ salt [9], several methods were proposed, including solar evaporation [10], adsorption [11,12,13,14,15], complexation [16,17,18,19], precipitation [20,21], solvent extraction [22,23] and membrane process [24,25,26,27,28]. Among these proposed techniques and despite the time consumption and environmental effects of solar evaporation, this remains the most used method for actual lithium salt production [29].

Thanks to their energy efficiency, application facility and ecological sustainability, membrane technologies have attracted wide interest for lithium procurement using different membrane processes such as Nanofiltration [30,31], Reverse Osmosis (RO) [32], Dialysis [33] and Electrodialysis (ED) [34,35]. For efficient extraction and optimal product quality, the used membranes need to exhibit a specific selectivity towards Li^+^ compared to other existent cations. 

This Li^+^ recovery occurs via its separation from coexisting cations, both bivalents (M^2+^: Mg^2+^ and Ca^2+^) and monovalents (M^+^: Na^+^ and K^+^). Li^+^/M^2+^ separation can be highly achieved by Nanofiltration (NF) [36] or Selective ED (SED) [37,38,39] using monovalent cationic exchanger membranes, which only allow the transport of monovalent cations while blocking the bivalents ones. As for Li^+^/M^+^ separation, it is a more complicated key step to perform because of their similar characteristics (electrical charge, size and mobility) [40]. Several papers have discussed the separation of monovalent ions using combined membrane approaches [41,42]. Tang et al. [42] studied the Li^+^ and K^+^ separation using nanoporous negatively charged track-etched membrane by NF. They succeeded in separating Li^+^ from K^+^ with a selectivity coefficient S(Li/K) = 70 and a Li^+^ flux of 0.014 mol·m^−2^·h^−1^.

ED represents a widespread separation technique. The use of electrical potential difference and ion-exchange membranes IEM (anionic AEM and cationic CEM) allows the separation of ionic species as a function of their charge and the membrane permselectivity [43]. This process appears more advantageous than diffusion dialysis (DD) and crossed ionic dialysis (CID) due to the use of electrical potential difference as the transport gradient, which permits more fast and charges-selective ionic transport [44].

Hoshino and Terai [45] synthesized organic membranes impregnated with ionic liquid (N-methyl-N-propylpiperidium bis (trifluoromethanesulfonyl) imide: PP13-TFSI) for Li isotopes separation (^6^Li and ^7^Li). This membrane was later used by Hoshino [34] for the recovery of Li^+^ from seawater by the ED process. Using this membrane, only Li^+^ can significantly migrate through the ionic liquid and concentrate at 5.94% in the cathodic compartment. The same author [46] used the previous membrane stabilized by a coating on both sides with SELEMION^TM^ CMV. He succeeded in separating Li^+^ over Na^+^ and K^+^ and concentrated it at 24.5% using ED process with an initial amount of Lithium in the feed solution of 170 ppb.

In our previous work [47], we prepared novel Lithium Composite Membranes LCMs based on the introduction of a Lithium Conductor Glass Ceramic (LICGC) powder, in an anion-exchange polymer (PECH-DABCO and PES-NH_2_) and a non-ionic surfactant (BRIJ76). The BRIJ76 was used to enhance the homogeneity of LICGC powder dispersion in the membrane matrix. The desirable selectivity of the prepared membranes was obtained by creating Li^+^ percolation pathways through interconnected LICGC particles. Many DD tests were performed to evaluate the Li^+^/Na^+^/K^+^ separation performances and selectivities: S(Li/Na) and S(Li/K). The results proved the high Li^+^ selectivity of synthetized LCMs compared to Na^+^ and K^+^. 

The best selective membrane composition was found to be 50.5 wt.% of LICGC, 25.5 wt.% of PECH-DABCO, 18 wt.% of PES-NH_2_ and 6 wt.% of BRIJ76. This membrane exhibits the highest selectivity coefficient S(Li/Na) = 376 when separating only Li^+^ + Na^+^ and S(Li/Na) = 278 and S(Li/K) = 364 when separating a mixture of the three cations Li^+^, Na^+^ and K^+^. The Li^+^ transport through this LCM was tested lately [48] under different dialysis conditions. It was found that Li^+^ concentration increase, treatment duration, neutral pH and Cl^-^ as co-ions improve the Li^+^ transport in DD when it is present alone in the feed compartment. 

For CID process, Li^+^ diffusion increases, and the optimal transport was found at 0.1 M of HCl as receiving solution. For Li^+^/Na^+^ separation, the effect of the feed concentration ratio [Na^+^]_F_/[Li^+^]_F_ on Li^+^ diffusion and membrane selectivity was tested by DD and CID processes. In both cases, it was found that this ratio positively affects membrane performances (recovered Li^+^ ratio and membrane selectivity S(Li/Na)), even at high Na^+^ and low Li^+^ concentrations. Reusability tests show that LCM remains selective towards Li^+^ after three cycles of use with a high selectivity coefficient. These satisfactory results confirm the suitability of LCM for Li^+^ extraction.

In the present study, we test the use of this LCM membrane for Li^+^ extraction using the electrodialysis process. We start with testing the effects of imposed current density using two types of ED cell. The first cell is composed of only two compartments (anode and cathode) separated by the LCM membrane, while the second type is a four-compartment cell (anode, AEM, feeding solution, LCM, receiving solution, AEM and cathode). Then, we test the effect of feeding concentrations ratio ([Na^+^]_F_/[Li^+^]_F_) on the Li^+^ recovery rate and membrane selectivity.

## 2. Materials and Methods

### 2.1. Materials

LiCl and NaCl salts and HCl solution were purchased from Sigma-Aldrich (Seven Hills, Australia). Deionized water was used for the solution’s preparation.

### 2.2. Membranes 

A previous study [47] conducted by our group allowed us to identify the composition of the membrane that has the best chemical and electrochemical properties to extract lithium from sodium and potassium containing solutions in a very selective way. This membrane is a Lithium Composite Membrane (LCM) composed of 50.5 wt.%, 25.5 wt.%, 18 wt.% and 6 wt.% of LICGC, PECH-DABCO, PES-NH_2_ and BRIJ76, respectively, and prepared using the blending technique as described in [47]. The principal purpose behind this composition is to prevent the rigidity of LICGC ceramic membranes and combine the significant Li^+^ selectivity of LICGC particles with the flexibility of organic anionic exchange polymer (PECH-DABCO and PES-NH_2_). 

This combination allows simultaneous Li^+^ and anion transport across the LCM. Li^+^ diffuses through interconnected LICGC particles, and anions are transported through the used polymer. Other cations are generally not allowed to transfer through the membrane polymer because of its anionic exchanger nature. Several characterizations were performed to ensure the LCM morphological homogeneity, thermal stability and mechanical properties. SEM pictures demonstrate the homogeneous dispersion of LICGC particles in the matrix building selective Li^+^ percolation pathways. 

The main physicochemical properties of LCM in Table 1 show that this membrane is relatively thin and has a particularly low conductivity compared to conventional ion-exchange membranes. The conductivity of the LCM is mainly provided by ion migration through the PECH-DABCO and PES-NH_2_ polymer, with a small proportion generated, when using Li^+^, by the interconnected LICGC particles. A water content of 11.3% generally provides a good level of ion mobility with a fairly high selectivity.

The Anion-Exchange Membranes (AEM) used in this work for constituting the four-compartment ED cell was a commercial AMX membrane. This was composed of a poly(styrene-co-divinylbenzene) (PS-DVB) copolymer containing quaternary ammonium functional sites and supported by a PVC film.

### 2.3. Electrodialysis Cells 

ED experiments were carried out using a small glass Hittorf cell composed of two or four compartments as presented in Figure 1. For the two-compartment cell, the anode and cathode compartments were separated by the LCM and filled, respectively, by the feed and the receiver solution. This cell has the advantage of being simple to implement and allows a low global electrical resistance. In this particular case, the operation is more similar to electrolysis than to electrodialysis. However, for simplicity of presentation, we will keep the name two-compartment electrodialysis to contrast with the four-compartment electrodialysis.

In the four-compartment cell, in addition to the feed and the receiver compartments, we added anodic and cathodic compartments, which played the role of protector compartments containing Na_2_SO_4_ (0.1 M) solution. Two anion-exchange membranes AMX were added in order to protect the LCM membrane from electrodes reactions. In both cases, the feed and receiver contained LiCl + NaCl mixtures at different compositions and a 0.1 M HCl solution, respectively. The volume of each compartment (anode, feeding, receiver and cathode) in both ED cells (two-compartment and four-compartment cell) was 50 mL.

These two solutions were stirred using a pumping system (3 L·min^−1^) in order to ensure their homogeneity and minimize diffusion boundary layers effects. An Agilent E3634A DC Power Supply was used as the power source. Two platinized titanium electrodes of 5 cm^2^ surface were placed at the cell extremities to provide the current injection. The sealing of these compartments as well as the positioning of the membranes were ensured by a system of two half-flanges made of phenoplast with high thermal and mechanical performances. The effective surface of the membrane was 4.15 cm^2^.

For membrane performance determination, we measured the recovered cations concentrations in the receiver compartment. Selectivity coefficients towards Li^+^ over Na^+^ S(Li/Na) were calculated using Equation (1), where [Li^+^]_R_, [Li^+^]_F_, [Na^+^]_R_ and [Na^+^]_F_ are molar concentrations of Li^+^ and Na^+^ in the receiver and feeding compartment, respectively.
(1)$$S\left(\mathrm{Li}/\mathrm{Na}\right)=\frac{\frac{{\left[{\mathrm{Li}}^{+}\right]}_{R}}{{\left[{\mathrm{Li}}^{+}\right]}_{F}}}{\frac{{\left[{\mathrm{Na}}^{+}\right]}_{R}}{{\left[{\mathrm{Na}}^{+}\right]}_{F}}}=\frac{{\left[{\mathrm{Li}}^{+}\right]}_{R}\times {\left[{\mathrm{Na}}^{+}\right]}_{F}}{{\left[{\mathrm{Li}}^{+}\right]}_{F}\times {\left[{\mathrm{Na}}^{+}\right]}_{R}}$$

The recovery rate RR was determined according to Equation (2), where [M^+^]_R_ and [M^+^]_F_ are the molar concentrations of M^+^ (Li^+^ or Na^+^) in the receiver and the feeding compartment, respectively.
(2)$$\mathrm{RR}\left(M^{+}\right)=\frac{{\left[M^{+}\right]}_{R}}{{\left[M^{+}\right]}_{F}}\times 100$$

### 2.4. Analyses

Li^+^ and Na^+^ concentrations in the receiving solution were measured using Ionic Chromatography (Metrohm 861 Advanced Compact IC).

## 3. Results and Discussion

In this study, each experiment was repeated three times, and we considered the average values with their measurement uncertainties.

### 3.1. Determination of the Limiting Current Density 

In order to preserve the efficiency of the electromembrane process and to prevent the dissociation of water molecules on LCM–solution interfaces, the imposed current density is required to be lower than the limiting current density (LCD). This LCD, which refers to the total depletion of the solute in the membrane adjacent layer, can be determined using current–voltage curves. Figure 2 presents the current density–voltage curve for 0.05 M NaCl + 0.05 M LiCl solution and LCM (A = 3.14 cm^2^) using Guillou’s cell and a setup as described in [49]. The obtained current density–voltage curve does not follow the typical trend of classical monopolar IEMs due to the unconventional/unusual nature and composition of the used LCMs.

The value of this LCD was found to be around 16.9 mA·cm^−2^. Thus, we chose the tested values of the imposed current below this limit current.

### 3.2. ED Using Two-Compartment Cell

In order to test the influence of the imposed current density on Li^+^ transport through the LCM and its selectivity towards Li^+^, ED experiments were performed for 4 h using a solution at equal concentrations [Li^+^]_F_ = [Na^+^]_F_ = 0.05 M. These experiments were performed in the ED two-compartment cell. The obtained results are summarized in Figure 3.

An increase of the current density from 0.5 to 12.0 mA·cm^−2^ improves the transport of both cations through the LCM. Hence, the recovered ionic concentration increased from 0.592 to 6.920 mmol·L^−1^ for Li^+^ (that is 11.7-times more) and from 0.019 to 0.875 mmol·L^−1^ for Na^+^ (that is 46.1 times more). The more rapid increase rate in the concentration of Na^+^ in the receiver, than that of Li^+^, leads to a significant reduction of membrane selectivity from 31.7 to 7.9 (that is four-times less). This difference in the transport rate of the two cations can be explained by a combination of numerous factors detailed here below.

The transport of the two cations Li^+^ and Na^+^ through the LCM membrane is carried out in two ways: (i) mainly by successive jumps between vacancy sites of the interconnected LICGC particles [50], (ii) and slightly by leakage through the polymer containing positively charged sites and the membrane imperfections [51,52] by diffusion (under concentration gradient) or by migration (under the applied current).

The transport of Li^+^ takes place preferably through the particles of LICGC, which have very high selectivity towards this cation. The Li^+^ leakage through the positively charged polymer and membrane imperfections remains possible but much less important than that of Na^+^ due to the difference in their hydrated radius and mobilities when they are in aqueous solutions, as shown in Table 2. However, this difference is amplified when it comes to a dense medium, such as a charged polymer. 

As a function of current density, Li^+^ transport continuously/correspondingly increases across the entire membrane section; it rises more rapidly through the interconnected LICGC particles and more slightly through the used polymer when compared to Na^+^ ones. It should be noted that, under the effect of the applied current, the Na^+^ ions can also compete with the Li^+^ ions through the LICGC particles; however, this mode of Na^+^ transport remains marginal under our operating conditions (a relatively weak current and a fairly dilute solution).

Furthermore, proton production at the anode compartment influences the separation performance of the LCM. These generated protons compete with other cations (Li^+^ and Na^+^), and they carry a portion of the electrical charge through the LCM. According to the data in Table 2, these H^+^ have a much higher mobility and diffusion coefficient than other cations; therefore, they can more easily pass through the different components of the LCM membrane: they can pass through the LICGC by Li^+^/H^+^ exchange (as Lithium Ions Sieves) [56] and through the used charged polymer by protons leakage [57]. 

Gases are also produced at the electrode surface (H_2_ in the cathode, O_2_ and especially Cl_2_ in the anode). Figure 4 summarizes all the transport phenomena that took place in a two-compartment ED operation for lithium recovery, in which all the produced compounds (especially the H^+^ ions and the Cl_2_ gas) are in direct contact with LCM leading so to a decrease of its performances. It is known that Cl_2_ gas causes alterations of the charged polymer, which can affect its physicochemical properties and its capacity to exclude co-ions (Na^+^ and Li^+^) as reported in the literature [58]. Thus, in order to avoid this Cl_2_-membrane undesirable interaction, and to maintain LCM properties, we considered the use of a four-compartment cell.

### 3.3. ED with Four-Compartment Cell

This cell allows us to isolate the anodic and cathode compartments and to avoid any contact of the membrane with the products of redox reactions. The isolation of these compartments is done by placing two anion exchange membranes as shown in Figure 5. ED experiments are carried out for 4 h using a feeding solution at [Li^+^]_F_ = [Na^+^]_F_ = 0.05 M. The obtained results are displayed in Figure 6.

By comparing Figure 3 and Figure 6, we can deduce an improvement in the LCM separation performances with the four-compartment cell. Indeed, by changing from a two-compartment cell to a four-compartment cell and at 0.5 mA·cm^−2^, the Li^+^ transport is improved by 5.7%, while the Na^+^ transport is reduced by 23.25%. This attenuation of Na^+^ transport across the LCM membrane and the enhancement of Li^+^ passage can be attributed to the partial inhibition of H^+^ transport by using the AMX. 

This decrease of H^+^ transport through the LCM leads to an increase in the total number of passed cations (Li^+^+Na^+^) by passing to a four-compartment one. Additionally, as it was shown earlier [48], the presence of protons affects the membrane performance on dialysis processes (DD and ICD) because of their high mobility, their high diffusion and their easy passage through the different components of the membrane matrix (the charged polymer and the LICGC particles).

The recovery rate evolution of the two cations as a function of the imposed current density using a four-compartment cell is illustrated in Figure 7. Using this configuration, we were able to extract 14.85% Li^+^ and only 0.98% Na^+^ with S(Li/Na) = 15.1 by imposing a current density of 12 mA·cm^−2^ for 4 h, whereas only 1.26% Li^+^ and 0.03% Na^+^ could be extracted with higher selectivity of the order of 43.6 by imposing a low current density of 0.5 mA·cm^−2^.

We followed the pH variations in the feed and receiver compartments for the two current densities 0.5 and 7.2 mA·cm^−2^. We observed that the increase of the current density significantly reduced the pH from 5.3 at the initial time to 2.2 after 4 h of ED for the feed and from 0.9 to 0.6 for the receiver. This expected acidification of the solutions can be attributed to the protons transport through the AMX protection membranes and the LCM selective membrane. Indeed, protons move in successive jumps from one water molecule to another, following the Grotthus diffusion mechanism from the anode compartment to the feed compartment through the AMX [59]. 

As mentioned earlier, once in the feed compartment, these protons intensively cross the LCM membrane and thus inhibit its performance. Despite the improvement in LCM separation performance from a two-compartment ED cell to a four-compartment ED cell, the Li^+^/Na^+^ separation efficiency remains relatively limited due to the intensive passage of H^+^ through the AMX and LCM. These limitations could be overcome by using anion exchange membranes with low proton leakage.

In order to test the LCM separation performances as a function of the [Na^+^]_F_/[Li^+^]_F_ concentration ratio, ED experiments were performed using the four-compartment cell and applying a current density of 0.5 mA·cm^−2^ for 4 h. We chose different concentrations and concentration ratios in the attempt to be comparable to those encountered in lithium brines. The results are reported in Table 3. Figure 8 gives the variations of recovered lithium and sodium concentrations, their recovery rates as well as the LCM selectivity coefficient vs. the [Na^+^]_F_/[Li^+^]_F_ concentration ratio, using log–log coordinates.

Figure 8 reveals that the recovered Li^+^ and Na^+^ concentrations decreased continuously when their concentrations also declined in the feed compartment. This was expected since both diffusion and electro-migration phenomena are proceeding in the same direction. Membrane selectivity remains important in favor of Li^+^ even if the Na^+^ concentration sometimes exceeds the Li^+^ one. 

Here, Li^+^ transport through LICGC percolation pathways is more significant than Na^+^ transfer mainly through the charged polymer, although its transfer through LICGC channels should not be neglected. Furthermore, we note that the Na^+^ recovery rate decreases as a function of its concentration in the feeding compartment. This fact proves that LCM remains a sufficiently impermeable barrier to Na^+^ co-ion passage.

When the Na^+^ concentration becomes significantly higher than that of Li^+^ in the feed, we notice a stabilization of the Na^+^ recovery rate, a slight decrease in the Li^+^ recovery rate and especially a reduction of membrane selectivity, which nevertheless remains at relatively high levels (almost 100). A competition between Li^+^ and Na^+^ passages takes place and seems to provide an optimum of selectivity at [Na^+^]_F_/[Li^+^]_F_ ratio of around 10. Beyond this ratio and at Na^+^ concentrations of about 2000 mg·L^−1^, the membrane tends to lose its significant selectivity due to an important leakage of Na^+^ through the charged polymer and, to a lesser extent, through the LICGC particles.

We also compared the selectivity values obtained using ED cell at 0.5 mA·cm^−2^ (this study) and Cross-Ionic Dialysis operation (CID) (previous work [48]). In both cases, we used an HCl solution in the receiver, a ratio [Na^+^]_F_/[Li^+^]_F_ of 20 and a 4 h treatment. Therefore by CID, we recovered 3.19 mg·L^−1^ of Li^+^ and 0.15 mg·L^−1^ of Na^+^ with S(Li/Na) = 421, while by ED we recovered 2.8-times more of Li^+^ (8.86 mg·L^−1^) and 15.9-times more of Na^+^ (2.38 mg·L^−1^) but with a 5.7-times smaller selectivity coefficient (S(Li/Na) = 74.5). Through this comparison, we can see that the imposed current has a more pronounced effect on the passage of Na^+^ than that of Li^+^ given its previously presented properties (R_h_, u and D).

Thus, we can deduce that ionic transport through LCM membrane by ED is due, in our case, to the combined effect between the low density of the imposed current and the ionic concentration gradient. All this allows us to explain the difference in the selectivity of LCM towards Li^+^ between DIC and ED, when comparing the findings of this paper with our previous one.

For Li^+^ recovery, especially using membrane technologies, the trade-off or the compromise between the permeability, extraction rate and membrane selectivity towards Li^+^ remains a challenge. A further crucial parameter for membrane technology and its applicability concerns energy consumption. Indeed, this latter needs to be as low as possible to achieve the best cost-effectiveness of the proposed process. Thus, the main goal of worldwide research in this field is finding the most appropriate solution/membrane that simultaneously ensures an optimum recovery rate with a maximum Li-selectivity and minimum energy consumption to achieve the highest efficiency and the purest product quality. 

Table 4 presents a comparison of selectivity coefficients and recovery rates between our LCM membrane and others membranes cited in the literature and used in the ED process. This table highlights the difficulty of finding a suitable membrane, especially at a higher Na concentration in the feed compartment, which provides both a good Li-recovery rate and high selectivity. The latter condition is certainly ensured by the IL-i-OM type membrane [34], despite its modest RR(Li^+^) (from 5.94% to 22.2%).

This membrane is an organic impregnated with high Li^+^ selective ionic liquid (PP13-TFSI), which permits Li^+^ migration through it. Our LCM membrane seems to exhibit comparable performances to those of IL-i-OM since it provides a RR(Li^+^) of around 10 and a selectivity coefficient of approximately 110. In our opinion, LCM remains one of the potentially effective membranes for selective Li^+^ recovery even in the presence of large amounts of Na^+^. 

## 4. Conclusions

Lithium-sodium separation by the ED process using a Lithium Composite Membrane (LCM) composed of 50.5 wt.% of LICGC, 25.5 wt.% of PECH-DABCO, 18 wt.% of PES-NH_2_ and 6 wt.% of BRIJ76 was the purpose of this study. We were interested both in determining the selectivity coefficient of this membrane towards Li^+^ (S(Li/Na)) and in measuring its recovery rate (RR(Li^+^) as a function of two parameters, which were the applied current density and the concentrations of these two cations in the feed compartment. ED operations took place in a two-compartment cell known to be simple to implement or in a four-compartment cell allowing the feed and receiver compartments to be isolated from the electrode compartments, thus, avoiding the telescoping of electrodes reactions with the phenomena studied.

In both ED cells, the increase of the current density led to a reduction in the selectivity coefficient; however, this reduction was less pronounced in the four-compartment cell thanks to its protection from redox reactions on the electrode surfaces and the partial decrease of generated proton transfer. We therefore maintained this cell for the rest of our work.

The effect of [Na^+^]_F_/[Li^+^]_F_ ratio in the feeding compartment on membrane selectivity was also tested at 0.5 mA·cm^−2^ for the current density. It was shown that, even at a relatively high concentration ratio, the membrane preserves its selectivity and separation performances towards Li^+^. For very high ratio values, this selectivity and recovery rate of Li^+^ decreases. For [Na^+^]_F_/[Li^+^]_F_ = 10 and after 4 h of ED operation, we successfully recovered 10% of Li^+^ with a high selectivity coefficient around 112. 

Based on these results, we can objectively estimate that the concept of this selective extraction of Li^+^ from a mixture, even concentrated in Na^+^, using an ED process was validated. However, additional studies will be carried out to investigate the effect of the feeding solution composition (coexisting cations (K^+^, Mg^2+^ and Ca^2+^) and anions (Cl^−^, OH^−^ and SO_4_^2−^)) on membrane performance.

## Figures and Tables

**Figure 1 membranes-12-00244-f001:**
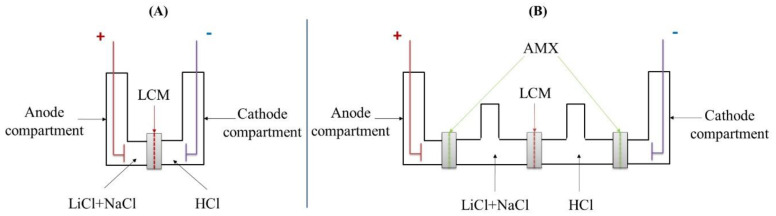
Schematic illustration of the ED cells with (**A**) two and (**B**) four compartments.

**Figure 2 membranes-12-00244-f002:**
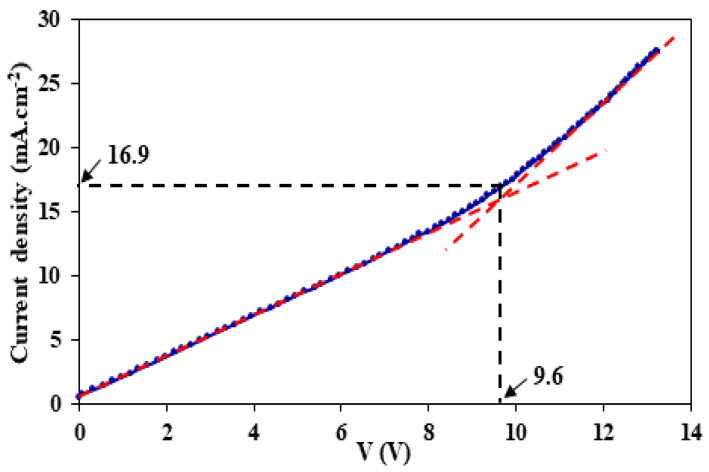
Current density–voltage curve for 0.05 M NaCl + 0.05 M LiCl solution and the LCM membrane.

**Figure 3 membranes-12-00244-f003:**
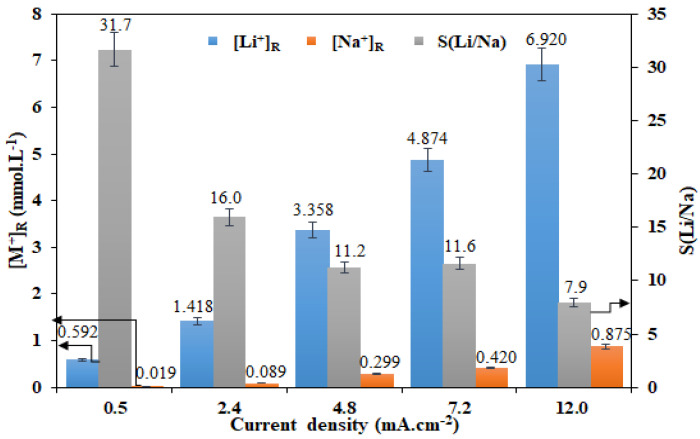
Effect of imposed current density on the Li^+^/Na^+^ separation performance of LCM using a two-compartment cell and for [Li^+^]_F_ = [Na^+^]_F_ = 0.05 M.

**Figure 4 membranes-12-00244-f004:**
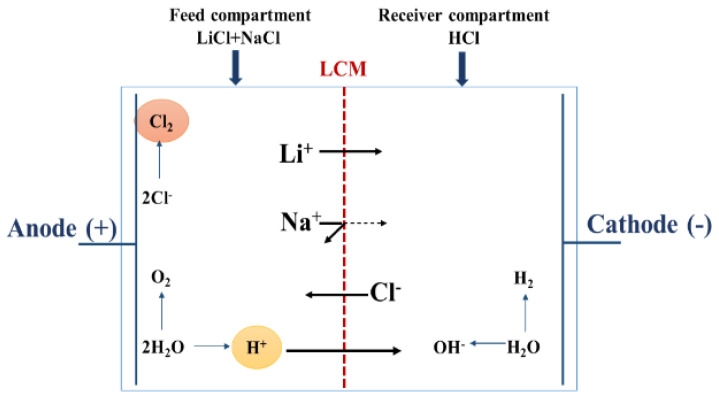
Schematic illustration and principle of a two-compartment ED cell.

**Figure 5 membranes-12-00244-f005:**
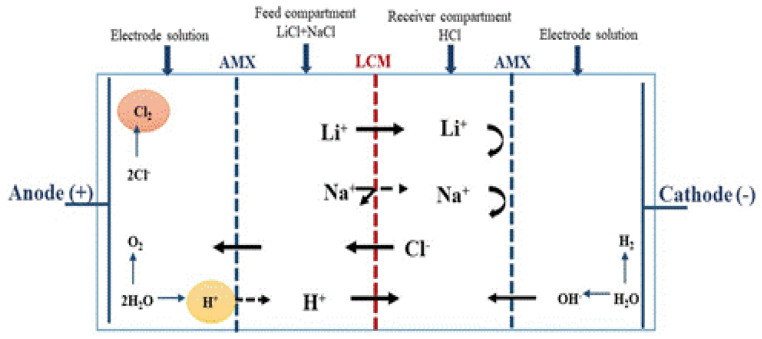
Schematic illustration and principle of four-compartment ED cell.

**Figure 6 membranes-12-00244-f006:**
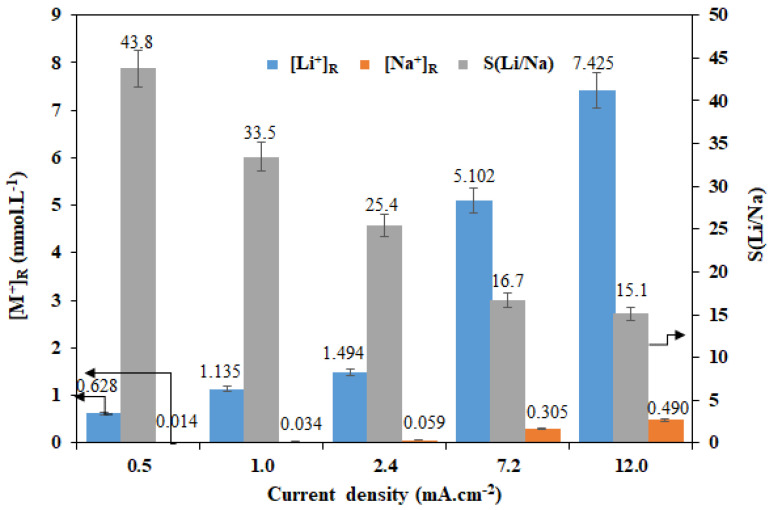
Effects- of the imposed current density on the Li^+^/Na^+^ separation performances of LCM using a four-compartment cell and for [Li^+^]_F_ = [Na^+^]_F_ = 0.05 M.

**Figure 7 membranes-12-00244-f007:**
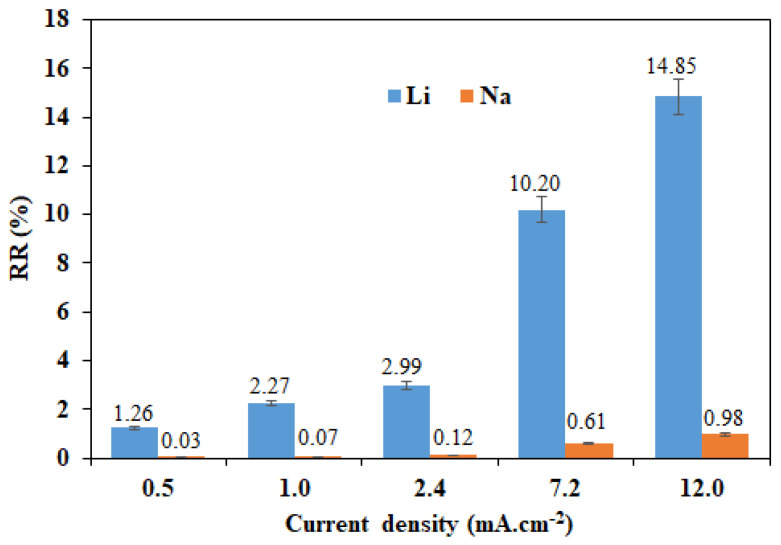
Li^+^ and Na^+^ recovery rates as a function of imposed current density using a four-compartment cell and for [Li^+^]_F_ = [Na^+^]_F_ = 0.05 M.

**Figure 8 membranes-12-00244-f008:**
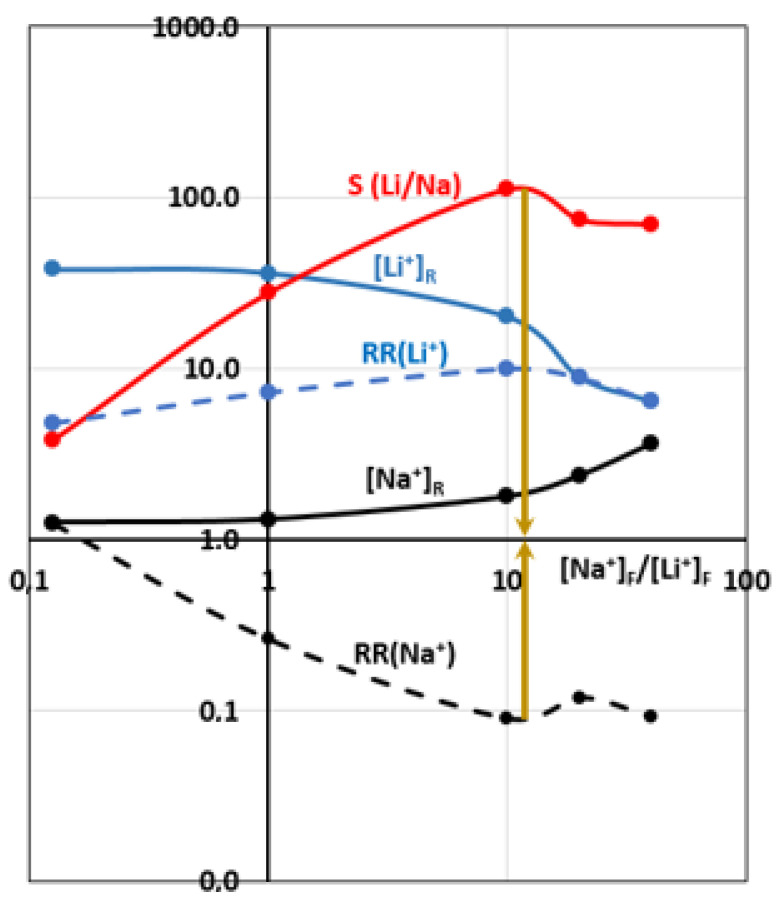
Recovered Li^+^ and Na^+^ concentrations, their recovery rates and LCM selectivity coefficient vs. the [Na^+^]_F_/[Li^+^]_F_ ratio, using log–log coordinates.

**Table 1 membranes-12-00244-t001:** Main static characteristics of the used LCM [47] and the AMX.

	LCM	AMX
Membrane thickness: l (µm)	130	153
Membrane conductivity (mS·cm^−1^)	0.75 in 0.1 M LiCl solution	12.6 in 0.1 M NaCl solution
Water content: W_c_ (%) *	11.3	24.8
Contact angle: θ (°)	61.3	63.0

*: $W_{c}=\frac{\left(W_{w}-{\;W}_{d}\right)}{W_{d}}\times 100,\;$ where W_w_ and W_d_ are the weights of the wet and the dry membrane, respectively.

**Table 2 membranes-12-00244-t002:** Characteristics of involved cations in aqueous solutions.

Cations	Hydrated Radius R_H_ (Å) [53]	Mobility u (10^−8^ m^2^·s^−1^·v^−1^) [54]	Diffusion Coefficients D (10^−9^ m^2^·s^−1^) [55]
H^+^	-	36.23	9.31
Na^+^	3.58	5.19	1.33
Li^+^	3.82	4.01	1.03

**Table 3 membranes-12-00244-t003:** Effect of [Na^+^]_F_/[Li^+^]_F_ ratio on LCM performances using a four-compartment cell at 0.5 mA·cm^−2^ for 4 h.

[Na^+^]_F_/[Li^+^]_F_	[Li^+^]_F_ (mg·L^−1^)	[Na^+^]_F_ (mg·L^−1^)	[Li^+^]_R_ (mg·L^−1^)	[Na^+^]_R_ (mg·L^−1^)	S(Li/Na)	RR(Li^+^) (%)	RR(Na^+^) (%)
0.125	800	100	38.14	1.26	3.8	4.77	1.26
1.0	500	500	36.29	1.31	27.7	7.26	0.26
10	200	2000	20.10	1.79	112.3	10.05	0.09
20	100	2000	8.86	2.38	74.5	8.86	0.12
40	100	4000	6.42	3.67	70.0	6.42	0.09

**Table 4 membranes-12-00244-t004:** Comparison between the ED performances of LCM and other reported membranes.

Membranes	Feed Composition	S(Li/Na)	RR(Li^+^) (%)	Reference
IL-i-OM membrane High durability IL-i-OM	[Li^+^]_F_ = 170 ppb [Na^+^]_F_ = 10,500 ppm	Very selective	5.94 22.2	[34]
PET track-etched membrane	* [K^+^]_F_ = 0.13 mol·L^−1^, [Li^+^]_F_ = 0.07 mol·L^−1^	0.20	-	[41]
Polymer inclusion membrane (PDT-PIM)	[Li^+^]_F_ = [Na^+^] _F_ = 20 mg·L^−1^	6.41	9.02	[60]
Sulfonated poly (ether ether ketone) composite CEM	[Li^+^]_F_ = [Na^+^]_F_ = [K^+^]_F_ = [Mg^2+^]_F_ = 500 ppm	2.17	84	[61]
CR67-MK111 (Homogenous polystyrene/Divinyl benzene)	[Li^+^]_F_ = 27,800 mg·L^−1^ [Na^+^]_F_ = 1350 mg·L^−1^	3.54	27.53	[62]
Monovalent- cation exchange membrane	[Na^+^]/[Li^+^] = 0.75 mol·L^−1^, [Li^+^] = 0.05 mol·L^−1^	1.25	21.47	[63]
Lithium selective cation exchange membrane	Feed: (LiOH·H2O = 1.9857 mol·L^−1^, NaOH = 0.0587 mol·L^−1^)	32.2	-	[35]
Supported liquid membrane based on a fluorinated molecule.	LiCl = NaCl = 15.10^−3^ mol·L^−1^	400	99	[64]
Monovalent selective ion exchange membrane	LiCl = NaCl = 0.05 mol·L^−1^	7.5	74.31	[65]
LCM	[Li^+^]_F_ = 200 mg·L^−1^ [Na^+^]_F_ = 2000 mg·L^−1^	112.3	10.05	This work

* Counter-current electromigration (combined ED and NF processes).

## Data Availability

Not applicable.

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
