# Peer review of "Lithium-Sodium Separation by a Lithium Composite Membrane Used in Electrodialysis Process: Concept Validation"

_membranes, 2022, doi:10.3390/membranes12020244_

Round 1

Reviewer 1 Report

This study investigated the lithium-sodium separation by an electrodialysis (ED) process using a Lithium Composite Membrane (LCM). The topic is worthy of investigation. But some remarks should be considered prior to be published on Membranes.

The authors said the used membrane in this paper is a Lithium Composite Membrane (LCM) composed of 50.5% wt., 25.5% wt., 18% wt. and 6% wt. But in table 1, only the properties of one kind of membrane was provided. Please specify the properties of the other membranes.

In Fig. 2a only one piece of membrane was used. The kind of electrochemical stack could not be called as electrodialysis but electrolysis process.

The ED experiments were performed in the ranged from 0.5 mA/cm2 to 12 mA/cm2. But as depicted in Fig. 2a, the limiting current density is around 16.9mA/cm2. Why the used current density is such lowered than the limiting current density? The authors finally concluded that 0.5 mA/cm2 was an optimal current density. This current density is extremely low; and the capital cost might be very high if this membrane was applied for the practical use. 

The RR of lithium in table 3 is so much low. Is possible to increase the RR by extending the experimental time?

In table 4, the authors compared the ED performances of LCM and other reported membranes. Please take in account the most pertinent and recent articles published for the comparison.

Reviewer 2 Report

The authors report "Lithium-Sodium Separation by a Lithium Composite Mem-brane used in Electrodialysis Process" through Concept Validation in the paper.

From the viewpoint of selective separation of Li-ion,  I think that this research can provide the possibility of highly selective recovery of Li-ion using the proposed four-compartment cell with minimizing the interfering effect of Na-ion in seawater by the specific ED process.

Therefore, I would like to recommend publishing this paper in this Journal.

Author Response

Comments from reviewer # 2 and our responses:

The authors report "Lithium-Sodium Separation by a Lithium Composite Membrane used in Electrodialysis Process" through Concept Validation in the paper.

From the viewpoint of selective separation of Li-ion, I think that this research can provide the possibility of highly selective recovery of Li-ion using the proposed four-compartment cell with minimizing the interfering effect of Na-ion in seawater by the specific ED process.

Therefore, I would like to recommend publishing this paper in this Journal.

Response: We thank you for your appreciation of our work, and for your very accurate reading of the problematics we are trying to solve.

Reviewer 3 Report

I recommend publishing the manuscript unchanged

Author Response

Comments from reviewer # 3 and our responses:

I recommend publishing the manuscript unchanged

Response: We thank you for your appreciation of our work, and for your very accurate reading of the problematics we are trying to solve.

Reviewer 4 Report

In short - the subject of research is very important, so the manuscript is worth publishing.

However, it needs to be corrected before publishing.

Here are some remarks and questions.

„The same authors [46]” - The same author [46].

„He succeeded in separating Li+ over Na+ and K+ and concentrated it at 24.5% by applying 2.0 V potential for 2 hours.” – 2 V and 2 h is useless information; how many electric charges were passed through the system? what was the initial amount of Li+?

„while the second one is a four-compartment cell (anode, feeding solution, receiving solution and cathode).” – the auxiliary membranes could be mentioned here.

„and has a relatively low conductivity compared to conventional ion-exchange membranes.” – 12.6/0.75 = ca.17 – 17 times lower conductivity cannot be described as „relatively low compared to”.

Membrane thickness: Tm - Tm usually denotes melting temperature; please, use another symbol, e.g. lm.

Water content: Wc (%) – how is it defined? with respect to the weight of wet or dry membrane?

„In both cases, the feed and receiver contain LiCl + NaCl mixtures at different compositions, and a 0.1 M HCl solution, respectively.” – what were their volumes? What were the volumes of solutions in the 2-comp. ED and in 4-comp. ED?

„Recovery rate RR was determined according to Eq. 2..” – question the same as above – what were the volumes of F and R?

„conductimetr” – conductometer

„In this study, each experiment is repeated…” – „Each experiment was repeated …”

Fig.2 – For monopolar ion-exchange membranes we usually obtain j = f(V) with plateau corresponding to limiting current density. Here this dependence is totally different. Please, explain it.

„An increase of the current density from 0.5 to 12.0 mA.cm-2 improves the transport of both cations through the LCM. Hence the recovered ionic concentration increased from 0.592 to 6.920 mmol.L-1 for Li+ (that is 11.7 times more) and from 0.019 to 0.875 mmol.L-1 for Na+ (that is 46.1 times more).” – How many moles of Li+, Na+ per mole of elementary charge (1 Faraday) passed through the system were transferred from F to R?

„(ii) and by leakage through the polymer containing positively charged sites” – in this case, cations are excluded from that area and the leakage should be small.

„when the density of the applied current increases, the Li+ transport through the LICGC particles increases very slightly,.. „ – Is it not proportional to the electric current?

Figures 3, 6, 7 should be changed – the current density should be on the x-axis, the type of plot should be changed from column to x-y. Then the real dependence of S and [M+]R on the current density will be visible.

To compare 2-comp. ED with 4-comp. ED the plots of S should be in the same figure. The same for [M+]R.

Table 2 – „Self-diffusion coefficients” – why „self-diffusion”?? In this table D and u are related by: u = D*F/RT.

„… by changing from a two-compartment cell to a four-compartment cell and at 0.5 mA.cm-2, the Li+ transport is improved by 5.7% while the Na+ transport is reduced by 23.25%. This attenuation of Na+ transport across the LCM membrane and the slight enhancement of Li+ passage can be attributed to the partial inhibition of H+ transport by using the AMX.” – I would expect a rather opposite effect. The presence of H+ means that a part of the electric charge is transported by H+. Thus, the transport numbers of Na+ and Li+ are decreased. The absence of H+ means the increase in the transport number of metal cations.

„We followed the pH variations in the feed and receiver compartments for the two current densities 0.5 and 7.2 mA.cm-2. We observed that the increase of the current density significantly reduces the pH from 5.3 at the initial time to 2.2 after 4 hours of ED for the feed and from 0.9 to 0.6 for the receiver.” – The receiver is separated from the cathode compartment by AMX (4-comp. ED). OH- anions from the cathode comp. freely move through AMX to the receiver. So why the decrease in pH is observed?

„A competition between Li+ and Na+ passages takes place and seems to provide an optimum of selectivity at [Na+]F/[Li+]F ratio of around 10. Beyond this ratio and at Na+ concentrations of about 2000 mg.L-1, the membrane tends to lose its significant selectivity due to an important leakage of Na+ through the charged polymer…” – For [Na+]F/[Li+]F < 10 the selectivity is low and [Na+]F is low, so the Na+ leakage is also low. How to explain the decrease in S for lower values of [Na+]F (Table 3)?

Table 4 – as I’ve mentioned above – the voltage values 2 V, 10 V, 7 V say nothing. One should give time, electric current (or moles of passed electric charge), volumes of solutions. The concentration units should be unified; the most adequate would be here mol/L or mmol/L.

English could be improved (but I’m not a native speaker so, maybe, I’m wrong).

Round 2

Reviewer 1 Report

The authors well reflected the comments.

Author Response

Dear Reviewer,

Thank you for your satisfaction with the revised version.

Reviewer 4 Report

„The pH values … were determined by a conductometer ..” – please, do not joke ;-)

------

Comment 4: Membrane thickness: Tm - Tm usually denotes melting temperature; please, use another symbol,
e.g. lm.
Response 4: In membrane science we usually use the notation (Tm). In order to prevent this confusion, we modified the notation to become (Thm).

No, in membrane science we do not use Tm as a membrane thickness, see e.g.:

1) RW Baker, Membrane Technology and Applications: symbol l,

2) Tanaka - Ion Exchange Membranes - Fundamentals and Applications: symbol d.

It is really important to use the most common symbols (in a given scientific discipline) – it makes the reading much easier.

---------

Response 5: Water content (Wc) …  - so this formula could be inserted into Table 1.

---------

Comment 10: Fig.2 – For monopolar ion-exchange membranes we usually obtain j = f(V) with plateau corresponding to limiting current density. Here this dependence is totally different. Please, explain it.
Response 10: J=f(v) curves depends on several parameters including membrane’s composition, homogeneity of its surface as well as solution’s composition of concentrate and diluate compartments on both side of membrane. We are agree with you that the typical j=f(v) shape of Ion-Exchange Membranes (IEMs) contain 3 regions with a plateau in the middle one. Here, our membrane is not an IEMs. We do not have a detailed and exact explanation of this curve. We prefer not to give information of which we are not sure.

But such a short comment could be given, i.e. „our membrane is not an IEMs”, therefore j = f(V) does not resemble the typical plot for monopolar IEMs, etc. Any reader who knows IEMs will notice this different shape of curve.

-----------------

Comment 16: Table 2 – “Self-diffusion coefficients” – why “self-diffusion”?? In this table D and u are related by: u = D*F/RT.
Response 16: Effectively, u and D are the mobility and self-diffusion coefficients of ions at infinite dilution in solution, respectively. u and D are correlated by the relation D=u*(RT/|z|F) with z=1 in our case. In fact, at infinite dilution, Self-diffusion coefficient of species/ions represents its diffusion coefficient in a single component system/single salt solution [*].

Yes, see the formula on the IUPAC page: https://goldbook.iupac.org/terms/view/S05582

But the values of u (and D = u*RT/F) in your table 2 are for the infinite dilution – there is no interactions between the same (or not the same) ions and the concept of „self” loses its meaning. So it’s enough to call D just "diffusion coeff.". If you don’t believe then use the frictional formalism of ion transport – see e.g.:

1) Laity, R. W. (1959). An Application of Irreversible Thermodynamics to the Study of Diffusion. The Journal of Physical Chemistry, 63(1), 80–83;

2) Laity, R. W. (1959). General Approach to the Study of Electrical Conductance and Its Relation to Mass Transport Phenomena. The Journal of Chemical Physics, 30(3), 682–691.

Author Response

Dear Reviewer,

Thank you for your efforts and constructive comments. Please find our point-by-point response to these comments.
